# How Has the Hazard to Humans of Microorganisms Found in Atmospheric Aerosol in the South of Western Siberia Changed over 10 Years?

**DOI:** 10.3390/ijerph17051651

**Published:** 2020-03-03

**Authors:** Alexandr Safatov, Irina Andreeva, Galina Buryak, Olesia Ohlopkova, Sergei Olkin, Larisa Puchkova, Irina Reznikova, Nadezda Solovyanova, Boris Belan, Mikhail Panchenko, Denis Simonenkov

**Affiliations:** 1Department of Biophysics and Ecological Researches, FBRI SRC VB “Vector” of Rospotrebnadzor, Koltsovo, 630559 Novosibirsk rgn., Russia; andreeva_is@vector.nsc.ru (I.A.); buryak@vector.nsc.ru (G.B.); ohlopkova_ov@vector.nsc.ru (O.O.); olkin@vector.nsc.ru (S.O.); puchkova@vector.nsc.ru (L.P.); reznikova@vector.nsc.ru (I.R.); solovyanova_na@vector.nsc.ru (N.S.); 2Laboratory of Atmosphere Composition Climatology, V.E. Zuev Institute Of Atmospheric Optics SB RAS, 634055 Tomsk, Russia; bbd@iao.ru (B.B.); pmv@iao.ru (M.P.); simon@iao.ru (D.S.)

**Keywords:** atmospheric aerosols, bioaerosols, culturable bacteria, long-term trends, hazard for human

## Abstract

One of the most important components of atmospheric aerosols are microorganisms. Therefore, it is necessary to assess the hazard to humans, both from individual microorganisms which are present in atmospheric bioaerosols as well as from their pool. An approach for determining the hazard of bacteria and yeasts found in atmospheric bioaerosols for humans has previously been proposed. The purpose of this paper is to compare our results for 2006–2008 with the results of studies obtained in 2012–2016 to identify changes in the characteristics of bioaerosols occurring over a decade in the south of Western Siberia. Experimental data on the growth, morphological and biochemical properties of bacteria and yeasts were determined for each isolate found in bioaerosol samples. The integral indices of the hazards of bacteria and yeast for humans were constructed for each isolate based on experimentally determined isolate characteristics according to the approach developed by authors in 2008. Data analysis of two datasets showed that hazard to humans of culturable microorganisms in the atmospheric aerosol in the south of Western Siberia has not changed significantly for 10 years (trends are undistinguishable from zero with a confidence level of more than 95%) despite a noticeable decrease in the average annual number of culturable microorganisms per cubic meter (6–10 times for 10 years).

## 1. Introduction

Human exposure to air pollution is among major health problems. Some pollutants can be easily monitored, doses obtained by individuals may be calculated and the results of such exposure may be predicted if the “dose–effect” dependencies are known. These dependencies are not known for most bioaerosols. An atmospheric bioaerosol is omnipresent part of atmospheric aerosols, accounting for up to 95% of their total number or up to 80% of their mass [1,2,3,4]. The peculiarity of atmospheric bioaerosols is, besides the usual effects of aerosols on atmospheric processes and climate [5,6,7], the ability to cause or provoke various infectious or non-infectious diseases in humans [5,8,9,10,11,12]. The most important component of atmospheric bioaerosols is microorganisms. Since microorganisms are usually hazardous to humans, it is important to be able to assess what this hazard is in each air sample. It is also important to be able to track the change in this hazard in the air of controlled points over time. Modern molecular biological methods can quite quickly reveal the biological diversity of microorganisms present in a sample [13,14,15,16,17,18,19,20,21,22,23,24,25,26,27], but for the vast majority of known microorganisms, their danger to humans has not been studied. Therefore, an approach that is based on growth, morphological and biochemical properties of culturable microorganisms (even if a microorganism has never been isolated from the natural environment early or most of its properties have not been studied) may be very useful.

We have previously proposed such an approach for determining the hazard of bacteria and yeasts found in atmospheric bioaerosols for humans and demonstrated its capabilities using the available experimental data as an example [28]. In the framework of the developed approach, a dimensionless integral index of the hazard to humans of microorganisms isolated from atmospheric bioaerosols was constructed, which is the product of four lower-level dimensionless integral indices characterizing the complex properties of the microorganism. One of them is an integral index of the concentration of culturable bacteria in an aerosol sample. Other components of the integral index of the hazard of microorganisms to humans are [28]:an index that evaluates the pathogenicity or hazard of individual isolates of microorganisms to humans;an index that evaluates the resistance of microorganisms to adverse environmental factors;an index that evaluates the resistance of microorganisms to antibiotics or other drugs.

Obviously, the more culturable microorganisms (identical or different) are in the air, the more dangerous they are to humans. The value normalized to the maximum number of culturable microorganisms in all samples studied (or in one sample; in this case, of course, this index is equal to 1) represents the first index [28]. The second index assesses the hazard (pathogenicity or conditional pathogenicity) of microorganisms to humans. Its normalized value reaches 1 for pathogens and is strictly equal to zero for completely non-pathogenic microorganisms. This index is based on the morphological and biochemical properties of microorganisms and allows us to predict their individual or collective pathogenicity in the range of 0 to 1. The third normalized index assesses the resistance of microbial cells to adverse environmental factors. It considers their growth, morphological and biochemical properties. The more resistant the microorganism is in the external environment, the greater the likelihood of it entering the human body and maintaining the ability to trigger negative reactions. Finally, the fourth index is determined by the resistance of microorganisms to the action of antibiotics or other drugs. The higher this resistance, the more difficult it is to overcome the negative effects of microorganisms in the body and, consequently, the higher the hazard of such a microorganism. The approach proposed in [28] was illustrated using experimental data obtained in 2006–2008.

Changes occurring in nature (climate change, atmospheric processes changes, changes of habitats areas of animals, insects and vegetation, changes in water systems [29,30,31,32,33,34]) should also be manifested in changes in the abundance and biodiversity of bioaerosols in the atmosphere.

Western Siberia is a region in which global climatic changes are clearly manifested: permafrost thawing, decrease in snow cover time, temperature increase, powerful greenhouse gas emissions [35,36,37,38,39,40,41,42,43,44,45]. At the same time, changes in the state of health of the population of the region caused by these climatic changes are poorly studied (see, for example, [37,46,47]). The method we developed allows us to assess the change in the hazard to humans of cultivated microorganisms located in atmospheric aerosols, and as a result, the influence of this factor, which undoubtedly affects the health of the population of the region. Thus, the aim of this work is to obtain new data on the hazard to humans in the context of ongoing climate change, both for individual microorganisms (only bacteria and yeasts in this paper), which are present in atmospheric bioaerosol, as well as their pool, and compare the results with the results obtained previously. This study includes two on-ground sampling sites where a person breathes directly and where the presence of microorganisms is largely determined by local sources of bioaerosols. In addition, microorganisms in the atmosphere at altitudes of up to 7 km were studied and its bioaerosol composition was largely determined by bioaerosols remote sources.

## 2. Materials and Methods

To ensure the comparability of the results obtained in 2006–2008 and in 2012–2016, materials and methods used were mainly the same as described in [28]. These materials and methods which were used in 2006–2008 and in 2012–2016 are described in details below, and their differences are highlighted in special Section 2.4.

### 2.1. Atmospheric Air Sampling

Sampling of atmospheric air was carried out at three points in the studied region: on the site of the FBRI SRC VB "Vector" of Rospotrebnadzor (Vector), 4 times a day (the sampling starts at 10:00, 16:00, 22:00 and 4:00 on the next day) in the middle of the month; in the Klyuchi village, once a season for 7 consecutive days (usually the sampling starts at 8:00–9:00); about 50 km south of Novosibirsk using the “Optik-E” laboratory (Aircraft, Figure 1 and Figure 2) on one of the last days of each month. A map showing the position of two sampling points and a typical flight trajectory of a laboratory airplane is shown in Figure 3. The “Optic-E” laboratory mounted on an Antonov-30 aircraft (Figure 2, see also [48,49,50,51]) for 2006–2008 session includes device for bioaerosol sampling and additional devices for registration of physical and chemical aerosol characteristics, meteorological conditions, etc. [48,52]. Isokinetic air sampling was performed through the special inlet outside the cabin (inserts in Figure 2 and Figure 3) [48,53] at cruising speed of aircraft approximately 360 km/hour. The operation of the air intake with reduced dynamic pressure is based on the principle of Venturi. Further, outboard air with a pressure equal to the air pressure inside the aircraft cabin was supplied to a stand with impingers (Figure 4), into which it was sampled for analysis of the presence and concentration of culturable microorganisms. It is obvious that both the device for outboard air intake and the tubes leading to the impinger are characterized by a certain percentage of losses of aerosol particles during their transportation. However, since both the intake air device and the length of the tubes are unchanged in all samples, these losses are the same for all samples. The samples were taken during daytime (usually the time of flight was in interval 12:00–15:00) over the Karakan pine forest (sees Figure 3) successively at eight altitudes: 7000, 5500, 4000, 3000, 2000, 1500, 1000, and 500 m [28]. To create sterile conditions, before each sampling or each flight, all incoming tubes were rinsed with ethanol.

Stainless steel impingers with a critical nozzle [28] (its analogue is described in [54]) were used for air sample collection. These devices (manufactured by JSC “Experimental-design bureau of biological precision engineering”, Kirishi, Russia) maintain a constant flow rate at a pressure differential of more than 4 × 10^4^ Pa of air through the device. An A-D1-04 pump (JSC “Kot”, St. Petersburg, Russia) was used to pump air samples through the impinger. Fifty milliliters of noncolored Hanks’ solution (SIGMA) was used as a sorbing fluid. Above-ground samples were taken at the flow rate of 50 ± 5 L/min for 30 minutes and altitude samples for 5 or 10 minutes (10-minute sampling was used during 2006–2008 and 5-minute sampling was used during 2012–2016) at each altitude. The retention efficiency of this impinger for aerosols of more than 0.3 μm exceeds 80% making up a constant value of 90% ± 15% for particles with a diameter of more than 2 μm.

### 2.2. Culturable Microorganisms’ Concentration

The concentrations of culturable microorganisms were determined by standard microbiological methods. Samples were seeded onto Petri dishes containing agarized media. LB [55] was used to detect saprophyte bacteria; depleted LB medium (diluted 1:10) was used to isolate microorganisms inhibited by the excess of organic substances, starch–ammoniac medium [56] was used to detect actinomyces; soil agar was used for soil microorganisms, and Sabouraud medium [56] was used for lower fungi and yeast. Successive sample dilutions were prepared when necessary. The seedings were incubated in a thermostat at a temperature of 28–30 °C for 3–14 days. Some isolates were additionally incubated at a temperature of 6–10 °C in 2012–2016. Phase contrast light microscopy was used for the study of morphological characteristics of bacteria (live cells and fixed Gram-stained ones too) and its colonies. Taxonomic groups the detected microorganisms referred to were determined according to [57,58,59]. Nucleotide sequences of PCR products corresponding to the fragments of 16S rDNA gene was performed for some bacteria [60,61]. The numbers of culturable microorganisms in samples were calculated according to standard methods [62]; the number of microorganisms was averaged over 3–4 parallels of samples 4–5 seeded on different media.

### 2.3. Microorganisms’ Biochemical and Morphological Characteristics

#### 2.3.1. Pathogenic Properties of the Isolates

Isolated microorganisms were tested for the presence of the following signs of pathogenicity:plasma-coagulase activity was determined by placing a loop of the test culture in a test tube with 0.5 mL of rabbit citrate plasma diluted with 0.9% sterile sodium chloride solution and incubating the suspension at 37 °C for up to 24 hours; a positive reaction was determined by plasma coagulation [28];hemolytic activity was determined by seeding cultures on nutrient agar containing 5% defibrinated blood and the seeding was incubated at 37 °C for 24–48 hours; a positive result was the presence of hemolysis zones around the grown colonies [59];fibrinolytic activity was determined by sowing 0.25 mL of an 18–20-hour broth culture of the test strain in test tubes containing 0.1 mL of citrate plasma, 0.4 mL of 0.9% sodium chloride, 0.25 mL of 0.25% CaCl_2_ solution; tubes with suspension were kept in an incubator at a temperature of 37 °C for 15–20 minutes; a clot formed in a test tube with a positive reaction (the presence of fibrinolysin) liquefies after the next two hours of incubation in a thermostat; the test culture does not have fibrinolytic properties if the clot persists, as in the control tube, where the culture was not added;gelatinase activity was determined by sowing microorganisms by injection in test tubes with meat–peptone broth containing 12% gelatin, kept in an incubator for up to 20 days; a liquefaction of the nutrient medium is noted in the presence of the gelatinase enzyme [57].

#### 2.3.2. Growth Characteristics of Bacteria at Increased Salt Concentration

The resistance of the studied microorganisms to high salt concentrations was determined when they were sown on a complete nutrient medium with the addition of NaCl at concentrations of 1%, 5% or 10%. After incubation at the optimum temperature, the range of resistance of the tested microorganism to the concentration of salts in the nutrient medium was established by the nature of growth or its absence.

#### 2.3.3. The Determination of Enzymatic Activity of Isolated Bacteria

Isolated microorganisms were tested for the presence of the following signs of enzymatic activities:proteolytic activity was determined by plating the studied microorganisms on an agar medium containing milk casein (milk agar); to prepare agar with casein, two components were prepared: 3% “hungry” agar (distilled water + 3% agar), sterilized at 1 atm for 30 minutes, and 12% milk sterilized at 1 atm for 20 minutes; then, agar was cooled to 50–55 °C, mixed with milk heated to the same temperature under aseptic conditions in a 1: 1 ratio, and poured into Petri dishes; after solidification of the medium, cultures were streaked and incubated under optimal conditions; the formation of transparent hydrolysis zones around crops in casein-containing agar indicated protease production [57];the amylolytic activity of cultures of microorganisms was determined when they were streaked on starch–ammonia agar; after incubation in a thermostat for 24–48 hours, the grown cultures in the dishes were poured into 5 mL of Lugol’s solution; the appearance of bright areas around crops within 3–5 minutes is a positive result;determination of lecithinase and lipase activity was carried out by two methods:seeding cultures with a stroke on yolk nutrient agar, for the preparation of which, under aseptic conditions, in yolk of molten and cooled to 50–55 °C fish-peptone agar (FPA) medium, yolk from a chicken egg is introduced; then medium is thoroughly mixed and poured into Petri dishes; the studied culture is streaked, incubated at the required temperature for 24–48 hours and the result is taken into account; when observed in an oblique light, lipase production was judged by the formation of a pearly shiny hydrolysis zone on agar around grown colonies; lecithinase (phospholipase) hydrolyzes the yolk lecithin; as a result of the precipitation reaction, a turbid whitish zone forms around the lecithin-fermenting colonies [57];plating cultures on a complete LB or RPA medium with 1% Tween-20 or Tween-40 and 0.01% CaCl_2_ as a substrate; sown cultures were incubated in a thermostat for 3–4 days and the result of the presence or absence of hydrolysis zones was determined [57];testing of cultures for the production of alkaline phosphatase was carried out using a reaction mixture of the composition: 0.3 mL of 0.85% NaCl added to 0.3 mL of substrate solution containing 0.04 M glycine buffer pH 10.5 and 0.01 M disodium-n-nitrophenyl phosphate (Sigma); the reaction mixture was incubated at 37 °C for 3 hours; Positive reaction manifested itself as yellow staining of the reaction mixture [59]; enzyme activity was determined within 3 hours of incubation by absorption on Uniplan apparatus (Russia) with a color filter at the wavelength of 450 nm.nuclease activity was determined by streaking cultures on RPA medium with the addition of an aqueous DNA solution to a final concentration of 2 mg/mL; before filling the medium, a sterile solution of CaCl_2_ (0.8 mg/mL) and 0.01% toluidine blue were added to the Petri dishes; in the case of the formation of DNase, a pink colored zone arose around the bacterial culture [59];the concentration of plasmid DNA in the strains was determined with screening method using a standard procedure; cells from a solid medium were suspended with a loop in 100 μl of buffer (50 mM Tris pH 8.0, 50 mM Na2-EDTA, 15% sucrose), 200 μl of alkaline solution (0.2 N NaOH, 1% SDS) and 150 μl of 3 M sodium acetate pH 5.0 were added, and centrifugation was performed for 5 minutes on a desktop centrifuge, then 1 mL of 96% ethanol was added to the sediment. The obtained DNA was analyzed in 0.8 % agarose in Tris-borate buffer pH 8.0 [48];when screening the strains for the presence of restriction endonucleases, individual colonies collected from a solid culture were suspended in 100–200 μl of TEN-buffer (0.1 M Tris, pH 7.5, 0.01 M EDTA, 0.05 M NaCl), lysocime and triton X-100 were used to destroy the cell wall of bacteria; the obtained cell extract was used for analysis for the presence of restriction endonucleases. DNAs of phages λcI857 and T7 were used as substrates for hydrolysis; electrophoresis of DNA after restriction was performed in 1% agarose (Sigma) [63]; the presence of restriction endonucleases in microorganism strains was revealed by the appearance of discrete fragments of substrate DNA in electrophoregram in UV light.

#### 2.3.4. Microorganisms’ Antibiotic Resistance

The sensitivity of microorganisms to antibiotics was determined by diffusion using the discs method [57]; the concentration of antibiotics in the discs used was: ampicillin (10 μg/disk), neomycin (30 μg/disk), benzyl-penicillin (100 U/disk), levomycetin (30 μg/disk), carbenicillin (100 μg/disk), canamycin (30 μg/disk), oleandomycin (15 μg/disk), rifampicin (5 μg/disk), streptomycin (30 μg/disk), polymxin (300 U), erythromycin (15 μg/disk), lincomycin (15 μg/disk), oxacillin (10 μg/disk), gentamycin (10 μg/disk), tetracycline (30 μg/disk), vancomycin (30 μg/disk), amikacin (30 μg/disk), netilmycin (30 μg/disk), monomycin (30 μg/disk). Note that lists of antibiotics used were not the same for 2006–2008 and for 2012–2016.

### 2.4. Changes in the Methods Used in the 2006–2008 and 2012–2016 Studies

A Tupolev-134 aircraft (Figure 3) was used for sounding of the atmosphere since 2011 [64]. The speed of this aircraft was maintained at the level of the Antonov-30 aircraft to maintain stable operation of the air intake. For sampling air containing microorganisms, the same impingers were used on both airplanes and in on-ground sampling; Figure 4. As noted above, the sampling time for airborne atmospheric sounding was 10 minutes for 2006–2008 and 5 minutes for 2012–2016.When analyzing samples by cultural methods in 2012–2016, in addition to the cultivation temperature of 28–30 °C, some isolates were additionally incubated at the temperature of 6–10 °C.And the last difference between the methods of samples analysis in the 2006–2008 and 2012–2016 investigations-the lists of antibiotics the sensitivity of isolates to which was determined are slightly different. For 2012–2016, the sensitivity of isolates was determined for a larger number of antibiotics, but this was done for not all identified isolates.

### 2.5. Data Analysis and Statistics

The initial data for all measured values for which lower-level integral indices were calculated using the formulas published in [28] and these integral indices themselves are given in Appendix A (Appendix A and List of description). The average values of the integral indices and their standard deviations from the mean are given in these tables for each year of observation. Additionally, the maximum values of the corresponding integral for each year are given.

In Appendix A, the integral index of hazard of bacteria and yeasts found in atmospheric bioaerosols for humans are calculated from the lower level integral indices for each of the isolates according to the procedure described in [28]. Furthermore, using the method of regression and ANOVA analysis (built-in Microsoft Excel software), a statistical analysis of the data was carried out. All analysis results were obtained at a reliability level of 95%.

## 3. Results

Based on characteristics of isolated microorganisms using the methods described above, four indices were calculated, which form the integral index of the hazard of microorganisms found out in atmospheric aerosols for humans. These indices were calculated both for each isolate and averaged for each year of research.

Let us compare the data presented in [28] with the results obtained in this research using the same methods. It should be noted that due to resource constraints for the company 2012–2016, not all the characteristics of the identified isolates were investigated, as it was done during the company 2006–2008. However, according to the authors’ opinion, the results obtained on a truncated set of characteristics quite clearly demonstrate the changes occurring in the pool of cultured microorganisms isolated from atmospheric aerosols in the south of Western Siberia.

### 3.1. Long-Term Trends of Average Annual Concentrations of Culturable Microorganisms in the Atmospheric Aerosol of the South of Western Siberia

Figure 5 shows the long-term trends of average annual concentrations of culturable microorganisms in the atmospheric aerosol of southern Western Siberia. In constructing this graph, we used the results published in [65] and our own data obtained after the publication of this paper.

Long-term observations of the concentrations of culturable microorganisms (bacteria, fungi and yeasts) in the atmosphere indicate that the average annual values of these concentrations have a pronounced tendency to decrease. In particular, from 2008 to 2016, this decrease is at least three times. 

It should be noted that, for the concentration of cultuarable microorganisms in atmospheric aerosols in the south of Western Siberia, the intra-annual variability was revealed [65]. Figure 6 presents the data for 2001–2016 values averaged for each month of years 2001–2016, normalized to the corresponding annual average. As can be seen from this graph, the differences between the maximum and minimum concentrations are on average up to three times. Naturally, for each sampling, the concentration of microorganisms depends not only on the revealed dependence, but also on the specific meteorological conditions, the effect of which on this concentration can be very significant.

It should be noted that the data used to construct the integral index characterizing the concentration of isolates in the air make up only part of the raw data used to construct Figure 5 and Figure 6. This is due to the fact that, to construct the integral index characterizing the concentration of specific isolates in the air, only data on the concentrations of bacteria and yeast, but not all culturable microorganisms, are used. To calculate the total concentration of culturable microorganisms in the air, data are used on all grown colonies in the seeded sample, but not all colonies succeed in obtaining isolates suitable for further studies, namely, isolate concentrations are used to construct the corresponding integral indices, and finally, Figure 5 and Figure 6 are constructed for data for 2001–2016, and data obtained for a smaller number of years were used to construct integral indices.

### 3.2. Sustainability of Culturable Microorganisms in the Environment

The results of determining the integral index of resistance of microorganisms to adverse environmental factors are summarized in Table 1 for annual mean values and standard deviations from them.

As follows from the analysis of the data presented in Table 1 and in Appendix A for the sustainability index of culturable microorganisms in the atmospheric aerosol of the south of Western Siberia, differently directed trends are observed for different sites of observation. Changes in the sustainability index of culturable microorganisms in an atmospheric aerosol in the south of Western Siberia do not exceed 1.1–1.75 times for 10 years of observations. However, given the fairly large scatter of data for each year of measurement relative to average values, these changes cannot be considered non-zero. 

There are those isolates among the isolates of bacteria and yeasts that have values of the microorganism sustainability index in the environment from 0 to 1 and the average annual values of the indexes are at the level of 0.2–0.45. Note that the maximum index values close to or equal to 1 exist for a small number of bacteria and yeasts, not in every sample, and not even in every year (Appendix A).

Thus, the obtained experimental data do not reliably reveal differences at a reliability level of 95% between the sustainability indexes of culturable microorganisms in the atmospheric aerosol of the south of Western Siberia determined for 2006–2008 and for 2012–2016. At a 95% reliability level, all three trends are significantly indistinguishable from zero.

### 3.3. Potential Pathogenicity of Culturable Microorganisms in the Atmospheric Aerosol of the South of Western Siberia

The results of determining the integral index of potential pathogenicity of pathogenicity of culturable bacteria and yeast for humans in an atmospheric aerosol in the south of Western Siberia are summarized in Table 2 for annual mean values and standard deviations from them.

As follows from the analysis of the data shown in Table 2 and in Appendix A, there is no noticeable positive trend in the index of the potential pathogenicity of cultivated microorganisms in the atmospheric aerosol for air samples taken at the site of Klyuchi only. This increase from 2006 to 2016 is 1.5 times. The obtained experimental data do not reliably reveal differences at a reliability level of 95% in indices of potential pathogenicity of culturable microorganisms in the atmospheric aerosol of the south of Western Siberia determined in 2006–2008 and in 2012–2016 for the Klyuchi sampling site. At a 95% reliability level, all three trends are significantly indistinguishable from zero.

The average annual level of this index for all isolates was 0.2–0.4, while only a few of the microorganisms detected over the entire period had indices of the potential pathogenicity of culturable microorganisms in the atmospheric aerosol, approaching but not reaching 1.

### 3.4. Culturable Microorganisms’ Resistance to Antibiotics

The results of determining the integral index of antibiotic resistance of culturable bacteria and yeast in an atmospheric aerosol in the south of Western Siberia are summarized in Table 3 for annual mean values and standard deviations from them.

The situation for this index is similar to the situation with the sustainability index of culturable microorganisms in the atmospheric aerosol of the south of Western Siberia. The trends of this index are multidirectional and over 8 years of observations of changes in the culturable microorganisms in the atmospheric aerosol in the south of Western Siberia resistance to the action of antibiotics do not exceed 1.5 times (Table 3 and Appendix A). Note that in 2015 and 2016, experimental studies on resistance to antibiotics of isolates from atmospheric aerosol have not been conducted. Given the fairly large scatter of data relative to the average values for each year of measurement, these changes can be considered indistinguishable from zero. Thus, the obtained experimental data do not reliably reveal differences at a reliability level of 95% between the indices of culturable microorganisms in the atmospheric aerosol in the south of Western Siberia resistance to antibiotics determined in 2006–2008 and in 2012–2014. At a 95% reliability level, all three trends are significantly indistinguishable from zero.

Among the identified isolates, there are both those that are susceptible and resistant to all studied antibiotics. It was found that the proportion of isolates sensitive to the action of all antibiotics or resistant to the action of no more than one antibiotic did not change significantly during the period studied. In 2006–2008, there were from 33% to 42% of such isolates, and in 2012–2014 there were from 28% to 42%. Two isolates, which are resistant to all studied antibiotics (11 out of 11) or 0.21%, were detected at all observation sites in 2006–2008 and in 2012–2014 there were three such isolates (8 out of 8, 9 out of 9, and 15 out of 15), or 0.95%, Appendix A. Consequently, the number of microorganisms that are most resistant to the action of antibiotics over 8 years has increased more than four times, while the indices of culturable microorganisms in the atmospheric aerosol in the south of Western Siberia resistance to the action of antibiotics have not changed.

The sources of atmospheric microorganisms, besides vegetation, animals, and human activity, are soil and water; most likely, bioaerosols should contain a similar percentage of bacteria resistant to the action of antibiotics. Indeed, about the same percentage of bacteria resistant to the action of at least one of the antibiotics studied was found for the water system [66,67,68,69,70,71] and in the soil [69,72].

### 3.5. Integral Index

The results of the determination of the integral index of hazard for humans of culturable bacteria and yeast in an atmospheric aerosol in the south of Western Siberia are summarized in Table 4 for annual mean values and standard deviations from them. 

As already noted, this integral index is the product of the 4 indices described above (see Appendix A). However, for some isolates of microorganisms, some indices were not determined, therefore, the average annual indices of this index for the corresponding year were substituted for the calculation. These values in the relevant cells of Appendix A are highlighted in light blue. In general, the average annual values of this index are not large and are in the range of 0.001–0.007. Recall that for individual pathogenic microbial isolates that are sustainable in the environment and do not possess antibiotic resistance, this index equals 1. Consequently, there are very few such pathogenic microorganisms in the atmospheric aerosol of the south of Western Siberia (see Appendix A). Moreover, the maximum value of this integral index over all the time of research for isolates from all observation sites is only 0.1074. At the same time, a fairly large number of isolates was found in the samples, for which the integral index is strictly equal to zero (see Appendix A). Here is interesting situation: when the concentration of one of the isolates is very high (normalized index about 1) other indices are low; and when, for example, pathogenicity index is high, other indices are low. Hence, the integral index is not large for all studied isolates. The dependence of integral index for monthly averages is pronounced for 2006–2008 when the number of isolates is large for many months [28]. The isolates number for 2012–2016 is low, they are found not in every sample. So, integral index for monthly averages is not representative in 2012–2016 (Appendix A for 2012–2016).

As follows from the data presented in Table 4, all long-term trends of these integral indices are unidirectional, showing a tendency to decrease; however, the magnitude of this decrease is almost indistinguishable from zero at a reliability level of 95%. Based on this, one can conclude that in 10 years of research, the hazard to humans of microorganisms isolated from atmospheric aerosols of the south of Western Siberia remains almost unchanged.

As it was mentioned above, the large scatter of the experimentally determined values included in this index (see Appendix A) makes it possible to speak only of the tendency of a small decrease in the integral index of the hazard of microorganisms isolated from atmospheric aerosols in the south of Western Siberia for humans. This decrease is primarily due to a decrease in the observed average annual number of culturable microorganisms in atmospheric aerosols in the south of Western Siberia. 

## 4. Discussion

As it was mentioned above, a long-term series of bacteria and yeast concentration observations have not been presented in the literature for other regions of the world; therefore, the results presented in Figure 5 refer only to the south of Western Siberia. A similar result is described in [73] for 2014–2018 for fungi yearly averaged concentrations in the air in France: the decrease is about 1.5 times. At the same time, an increase in the average annual concentrations of fungi of the genera *Aspergillus* and *Penicillium* in Derby, UK, in 1970–2003 was recorded in [74]. As for the intra-annual dynamics of changes in the concentrations of bacteria and yeast, there is information in the literature, although such long-term observations that are presented in this paper are also absent in the literature. It should be noted that for different microorganisms (strains, genera, kingdoms), these dynamics can vary significantly. In particular, in many cases, the concentration of culturable bacteria in a warm season is higher than in a cold one [75,76,77,78,79,80,81,82,83,84,85,86], but there are also opposite examples [80,87,88,89,90,91].

The noted decrease in the average annual concentrations of culturable microorganisms in the south of Western Siberia is probably associated with the ongoing climate changes, which are manifested in a change in the location and power of bioaerosol sources, pathways of predominant transport of bioaerosols in the atmosphere, etc.

There is also no information in the literature about the change over time of microorganism characteristics that are included in the index of potential pathogenicity of culturable microorganisms in the atmospheric aerosol and in the index of their sustainability in the atmospheric aerosol to adverse environmental factors. Similar comprehensive studies have not been conducted previously.

With regard to antibiotic resistance, the literature presents data indicating that there is a tendency to an increase in the number of bacteria resistant to the action of antibiotics over time, see, for example, [92,93,94,95,96]. In the collections of bacteria isolates collected over a long period of time, the latest trend can be traced reliably [97,98]. Moreover, more and more isolates with multiple antibiotic resistances are detected in nature [99,100,101,102].

It should be noted that while for ground-based observation points, the obtained data directly provide an assessment of the hazard to humans of microorganisms in the air. Let us remind the reader that a large portion of aerosol particles in the on-ground atmospheric layer originated from local sources. It is difficult to relate the results to any particular place on the Earth’s surface for high-altitude observations. Bioaerosols found in high-altitude samples of atmospheric air originated mainly from long-distance sources and can, due to the stochastic nature of their movement in the atmosphere, reach the surface at completely different points. In addition, during the transfer process, some microorganisms can simply be inactivated, or at least lose their ability to grow under cultivation conditions.

It should be noted that it is not easy to build the dynamics of the monthly change of all the above-mentioned indices because of their very high variability and a small number of identified microorganisms for some months. Very weak dependencies can be observed only.

All the results presented in Section 3 have very high variability for both the average annual values of integral indices and the values of these indices for individual isolates. This is due to the properties of an atmospheric aerosol, which is a complex mixture of particles of various compositions. These particles enter the atmosphere from various terrestrial sources of inorganic, organic or biological nature, and are also formed in the atmosphere during nucleation processes. While in the aerosol, particles are continuously transformed under the influence of changing temperature and humidity, due to the presence of volatile compounds in it, capable of entering into chemical interactions with particles under the influence of solar radiation, etc. These factors can lead to inactivation of microorganisms in the particles. It is also necessary to consider the processes of particle deposition (sedimentation, washout by precipitations). Because of this, the concentration and composition of an aerosol in the atmosphere can vary greatly in samples that are close in time or in distance. For example, in [65], it was mentioned that two air samples taken during aircraft sounding of the atmosphere (a difference of 1000 m in height and 15 minutes in time) contain radically different compositions of microorganisms. This also leads to the fact that, if the average annual dependencies of the total concentration of microorganisms shown in Figure 5 have a relatively small dispersion, the attempt to construct similar dependencies for individual phylums and lower taxonomic groups was unsuccessful due to the extremely large concentration dispersion [103]. All of the above is particularly pronounced in Siberia with its sharply continental climate, many local sources of bioaerosols (forests, meadows, swamps, industrial production) and the prevailing direction of the winds, which bring a lot of dust (and soil microorganisms to it) from north-western Kazakhstan to Western Siberia. Moreover, the authors are not sure that an increase in the number of samples analyzed for each month (season) will noticeably affect a decrease in the variance of the result and, therefore, an increase in the reliability of the dependencies identified in this article.

The approach proposed in [28] made it possible, for the first time, to quantitatively compare the potential danger to humans of the totality of microorganisms in various samples of atmospheric air. Using this unique approach, it was possible to assess the change in potential danger to humans over 10 years of observation at different points in Western Siberia. Currently, the authors are not aware of any other approaches that allow such estimates.

At the same time, the authors do not pretend to the universality of the method used in this article for assessing the hazard to humans of culturable bacteria and fungi located in atmospheric aerosol. The algorithm for estimating integral indices proposed in [28] is not exhaustive and, probably, there are still some characteristics of cultivated microorganisms (morphological, cultural, biochemical, and biophysical) that refine the constructed integral indices. In addition, the weight values of the various characteristics of microorganisms by which integral indicators are calculated may also need to be clarified, and finally, an approach similar to that proposed for bacteria and yeast was not developed for micromycetes. Therefore, the results obtained in this article reflect the hazard to humans of only bacteria and yeast present in the aerosol, but not for all culturable microorganisms. In addition, not all microorganisms are easily cultivated under standard conditions. The method used simply cannot be implemented for unculturable microorganisms.

## 5. Conclusions

Summing up the conducted research, one must conclude that, despite a noticeable decrease in the average annual number of culturable microorganisms in the atmospheric aerosol in the south of Western Siberia, the hazard of these bacteria and yeasts to humans has not practically changed for 10 years. Thus, neither passing climatic changes in the world, nor changes in the antibiotic resistance and sustainability of culturable microorganisms in the environment have affected the hazard to humans of bacteria and yeasts found in the atmospheric aerosol in the south of Western Siberia. 

The authors published in 2012 [28] an approach on how to estimate the hazard of bacteria and yeasts in atmospheric aerosols to humans, but no similar investigation has been conducted elsewhere in the world. Usually, the data of experiments are given in the literature, in which the effect of a specific bioaerosol that has penetrated the respiratory tract is studied, see, for example [104]. It is clear that such an approach is difficult to use to characterize a large number of bioaerosol samples. The authors hope that very important studies on risks of human exposure to microorganisms in the air will be estimated using our approach [28] and will be published, not only for Western Siberia; the authors also hope that a longer study of the properties of culturable microorganisms in atmospheric aerosols will reliably confirm the identified weak dependencies in the future in the south of Western Siberia.

## Figures and Tables

**Figure 1 ijerph-17-01651-f001:**
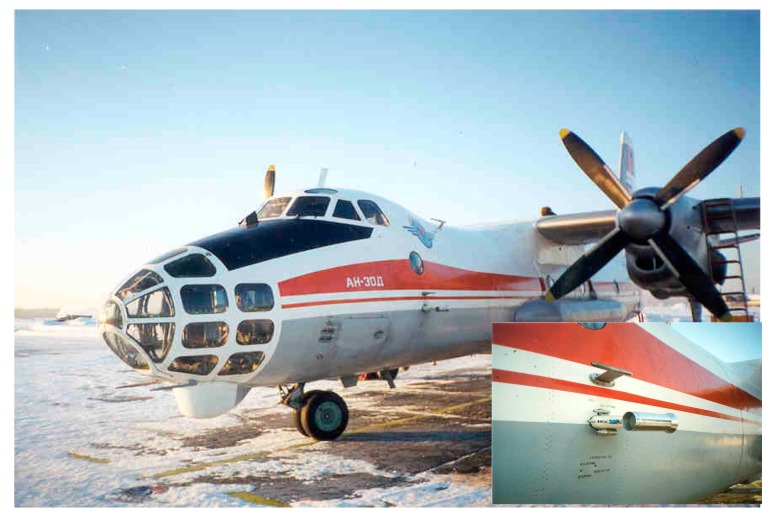
Photo of Antonov-30 aircraft. Insert is the samplers’ inlets.

**Figure 2 ijerph-17-01651-f002:**
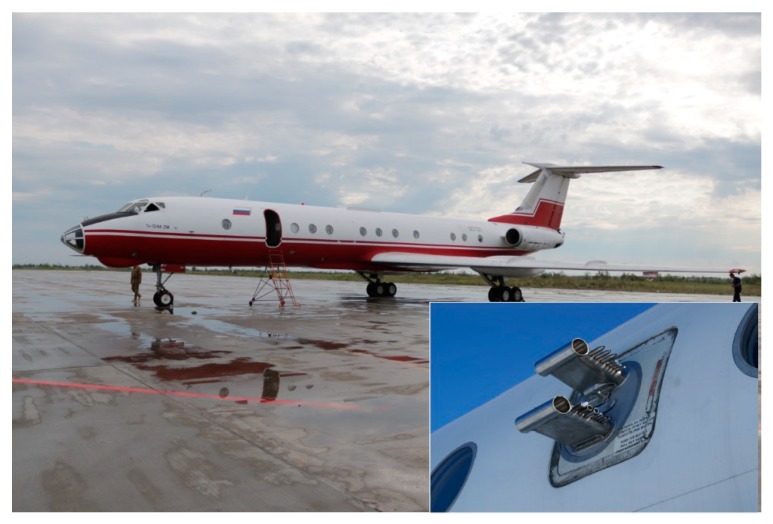
Photo of Tupolev-134 aircraft. Insert is the samplers’ Inlets.

**Figure 3 ijerph-17-01651-f003:**
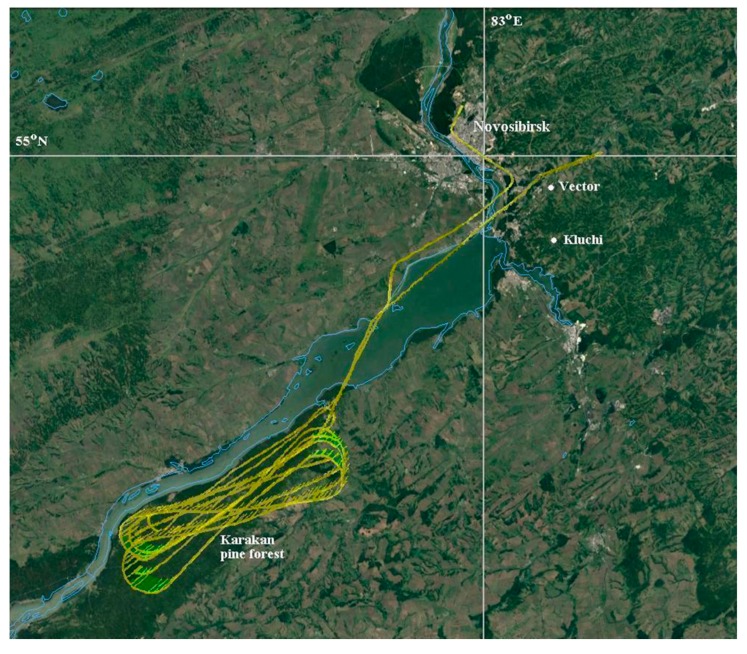
The map of sampling sites and aircraft’s flight typical trajectory.

**Figure 4 ijerph-17-01651-f004:**
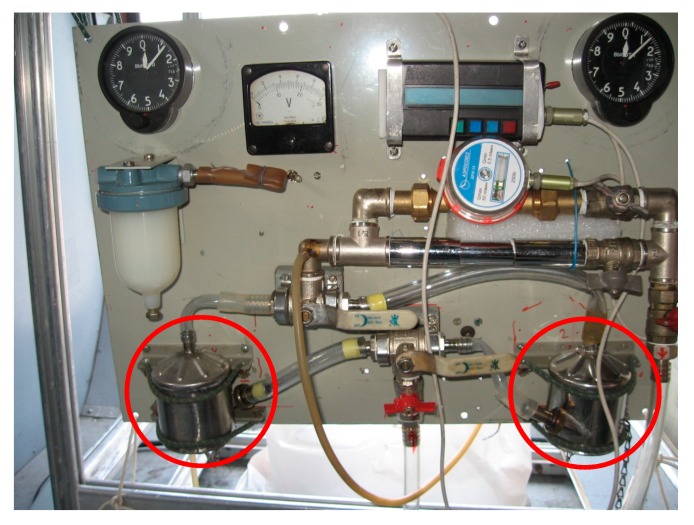
Photo of two impingers (red circles) at a sampling stand of microorganisms located in atmospheric aerosol.

**Figure 5 ijerph-17-01651-f005:**
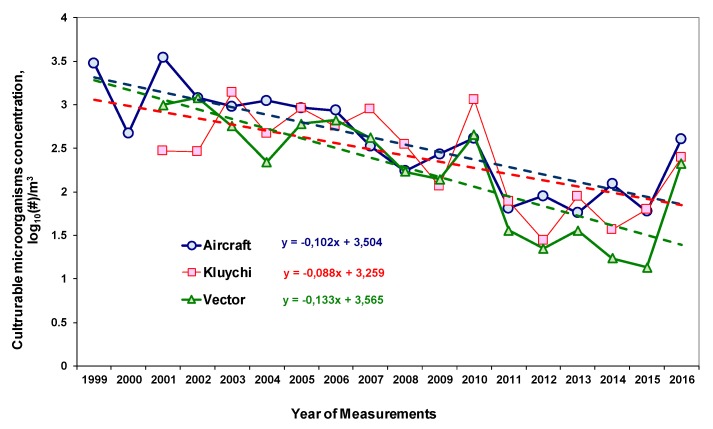
Trends of average annual concentrations of culturable microorganisms in the atmospheric aerosol in the south of Western Siberia. The average annual values with root mean square deviations (standard deviations) from the mean are given.

**Figure 6 ijerph-17-01651-f006:**
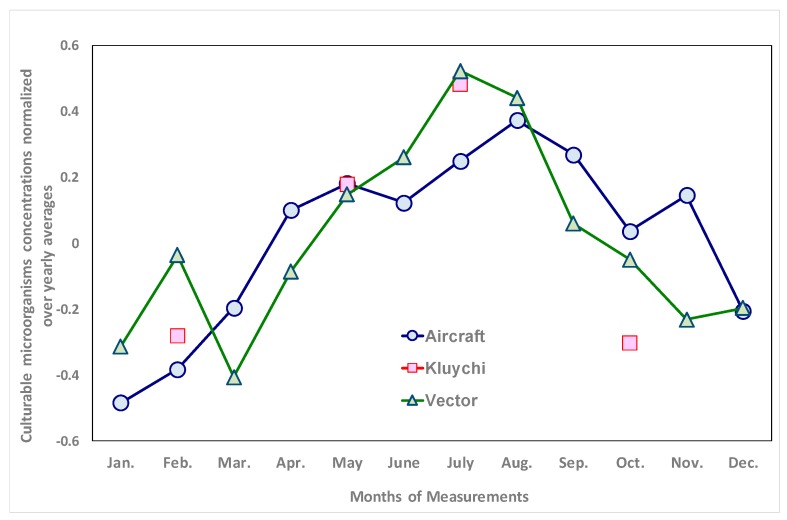
Intra-annual variability of concentrations of culturable microorganisms in the atmospheric aerosol of the south of Western Siberia. The average values for a given month for the entire observation period and the standard deviations from the average are given.

**Table 1 ijerph-17-01651-t001:** The annual average values and standard deviations from them of the integral index of the microorganisms’ resistance to adverse environmental factors values.

Sampling Site	Values	2006–2007	2007–2008	2012	2013	2014	2015	2016
Vector	Average	0.359	0.372	0.324	0.247	0.244	0.135	0.252
	Standard deviation	0.223	0.206	0.256	0.243	0.229	0.132	0.209
Klyuchi	Average	0.315	0.335	0.452	0.399	0.381	0.354	0.354
	Standard deviation	0.216	0.182	0.246	0.222	0.215	0.108	0.149
Aircraft	Average	0.307	0.399	0.567	0.380	0.415	0.405	0.431
	Standard deviation	0.212	0.222	0.333	0.191	0.137	0.125	0.176

**Table 2 ijerph-17-01651-t002:** Annual average values and standard deviations from them of the integral index of potential pathogenicity of culturable bacteria and yeast in an atmospheric aerosol in the south of Western Siberia.

Sampling Site	Values	2006–2007	2007–2008	2012	2013	2014	2015	2016
Vector	Average	0.239	0.239	0.315	0.202	0.213	0.374	0.292
	Standard deviation	0.148	0.121	0.177	0.117	0.146	0.177	0.175
Klyuchi	Average	0.209	0.275	0.232	0.205	0.210	0.371	0.175
	Standard deviation	0.132	0.139	0.187	0.138	0.134	0.197	0.175
Aircraft	Average	0.235	0.270	0.306	0.247	0.203	0.409	0.303
	Standard deviation	0.139	0.150	0.196	0.130	0.116	0.182	0.177

**Table 3 ijerph-17-01651-t003:** Annual average values and standard deviations from them of the integral index of culturable microorganisms ’resistance to antibiotics bacteria and yeast in an atmospheric aerosol in the south of Western Siberia.

Sampling Site	Values	2006–2007	2007–2008	2012	2013	2014
Vector	Average	0.207	0.206	0.170	0.195	0.152
	Standard deviation	0.181	0.129	0.173	0.208	0.107
Klyuchi	Average	0.262	0.245	0.265	0.239	0.144
	Standard deviation	0.211	0.146	0.106	0.231	0.107
Aircraft	Average	0.232	0.206	0.200	0.314	0.278
	Standard deviation	0.184	0.149	0.188	0.319	0.207

**Table 4 ijerph-17-01651-t004:** The annual average values and standard deviations from them of the integral index of hazard of culturable bacteria and yeast in an atmospheric aerosol in the south of Western Siberia.

Sampling Site	Values	2006–2007	2007–2008	2012	2013	2014	2015	2016
Vector	Average	0.0035	0.0007	0.0001	0.0011	0.0003	0.0001	0.0018
	Standard deviation	0.0110	0.0017	0.0002	0.0026	0.0008	0.00015	0.0061
Klyuchi	Average	0.0010	0.0047	0.0002	0.0005	0.0001	0.0007	0.0032
	Standard deviation	0.0045	0.0109	0.0003	0.0009	0.0002	0.0012	0.0057
Aircraft	Average	0.0016	0.0019	0.0000	0.0000	0.0037	0.0004	0.0018
	Standard deviation	0.0034	0.0058	0.0001	0.0001	0.0059	0.0004	0.0024

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
