# Peer review of "How Has the Hazard to Humans of Microorganisms Found in Atmospheric Aerosol in the South of Western Siberia Changed over 10 Years?"

_ijerph, 2020, doi:10.3390/ijerph17051651_

Round 1
Reviewer 1 Report
In general, the manuscript topic is interesting, however the paper to far too raw for publishing. I suggest using following comments to improve the quality of the manuscript.
The title is misleading; must be something related to comparison of the results from 2006 and 2016 investigations. Just reference to [42] regarding to similarity between impingers is not sufficient. There are many different types of such devices and I would like to see much more technical details along with verified outcomes of previous usage. Alternatively, results of some feasibility laboratory study would be important. The authors are suggested to pay more attention at the stage of submission as Figures 3 and 4 are not readable in the file submitted I suggest the authors to seriously reconsider their strategy of data presentation. R^2 values like 0.0015 or 2E-5 do not look very scientific causing a lot of questions about the results and outcomes. In general, I am not sure how any conclusions could be made considering huge scattering of the results. I understand that the nature of measurements does not ensure low discrepancy, however I would like to see number of repeats, STD, statistical tests (for ex ANOVA) or anything else enabling one to judge the manuscript outcomes. Finally, I do not think that publishing of 4-year-old results is a very good practice. Do you have something more recent? Grammar polishing is also recommended
Author Response
In general, the manuscript topic is interesting, however the paper to far too raw for publishing. I suggest using following comments to improve the quality of the manuscript.
The title is misleading; must be something related to comparison of the results from 2006 and 2016 investigations.
The paper’s title was changed (lines 2 – 4).
Just reference to [42] regarding to similarity between impingers is not sufficient. There are many different types of such devices and I would like to see much more technical details along with verified outcomes of previous usage. Alternatively, results of some feasibility laboratory study would be important.
The information on the impingers used and its calibration is presented in text (lines 131 – 141). The Photo of the sampling stand with 2 impingers is added also (Fig. 4).
The authors are suggested to pay more attention at the stage of submission as Figures 3 and 4 are not readable in the file submitted I suggest the authors to seriously reconsider their strategy of data presentation. R^2 values like 0.0015 or 2E-5 do not look very scientific causing a lot of questions about the results and outcomes.
Picture 5 and 6 in corrected version (3 and 4 in previous version) of the paper are improved. R2 is omitted. Observed trend is discussed (lines 292, 304-312).
In general, I am not sure how any conclusions could be made considering huge scattering of the results. I understand that the nature of measurements does not ensure low discrepancy, however I would like to see number of repeats, STD, statistical tests (for ex ANOVA) or anything else enabling one to judge the manuscript outcomes.
Special section on data analysis (2.5) is added to the text. Table S13 includes results of data analysis now.
Finally, I do not think that publishing of 4-year-old results is a very good practice. Do you have something more recent?
Now we have some data for 2019, but it is not complete yet. The investigation will be continued.
Grammar polishing is also recommended.
Authors understand that even after correction English is not ideal. So, we are ready to present our paper for English polishing to Publisher. We understand that any work deserves fee. And we will be ready to pay for it.
Reviewer 2 Report
This is an important and novel contribution on long-term monitoring of bioaerosols in Siberia and relevant human health impacts. The subject as well as the study area are worth of investigation and hence the work deserves scientific merit. However, there are several issues that need to be resolved before publication, as follows:
Title: Based on many of the comments below, the authors need to change the title, taking care of terms like "long-term" (this usually refers to trends, sometimes of decadal scale), like "atmospheric bioaerosol" (see below), and "impacts on human health" (they do not really, this is only at a discussion level). I would recommend to completely change the phrasing.
line 17: please omit 'atmospheric': bioaerosols are by definition atmospheric. Please change throughout the manuscript.
lines 18-19: not correct syntax, please edit.
lines 21-23: as written here, it reads as if previously published results (of biomonitoring?) are compared against recently acquired experimental data, so it is immediately not clear whether indeed the 2 datasets are comparable. If indeed they are identical, please rephrase to something like 'a similar dataset was acquired using the exactly same methods as in... so as to check long-term alterations of/in...'.
lines 25-26: please avoid using citations within the Abstract. Instead, use expressions like 'compared to previously published results by the authors in 2008', or similar.
line 26: "set of indices" is too vague for the abstract. Please provide more details or name the most important.
lines 28-30: there is an antiphasis here: "noticeable decrease" but "not changed significantly". Please provide statistical measures of both results, or change the phrasing.
Overall comment in the Introduction: the Introduction is somewhat too technical, based on previously published results (and methods) by the authors. Before that, and so as to keep a balanced proportion of relevant information on the topic, i would expect additionally the state of the art on: A) monitoring of bioaerosols, with an emphasis at a 2nd stage on the selected microorganisms in this paper, B) previously published studies on environments like Siberia (extreme environmental regime, more sensitive to environmental stress), climate change effects, human health effects (and what kind). After those, the authors would make sense to highlight the novelty of their work: how the current research advances the scientific knowledge. My suggestion would be to focus more on the study area: Siberia alone is a great (and not that much published) environment, with higher sensitivity because of climate, latitude and extreme conditions. Discuss all accordingly.
Methods, overall comment: the authors provide an extensive description of the processing/lab methods. However, the collection method is not proportionately described. Even if there is a previously published study, it is meaningful to briefly provide the concept of airplane measurements, as it is a very novel (and promising) biomonitoring tool. Please also add relevant information in the Introduction too. Moreover, add in Methods a more detailed description of the altitudes and the whole experimental design of the flight. A map would certainly significantly add up to the paper quality.
line 87: the differences are not really highlighted: if the reader has to go through all the methodological text to find these, it is confusing. Please create a dedicated sub-section highlighting potential differences, so as to make comparability clear and also ensure the soundness of the paper methods.
line 97: the devices used seem appropriate for this kind of sampling, but still, how have the authors ensured that all particles have been reliably and non-biased sampled under such a huge speed? Please discuss accordingly.
line 98: was the hour of the sampling taken into account? There is known diurnal periodicity of the bioaerosol abundance. Please discuss accordingly.
line 98: how were the different altitudes taken into account? The sample was pooled across all elevations? A map would elucidate the issue.
lines 110-113: i totally understand the unavoidable variability of all measurements, especially with such a monitoring device, like an airplane. However, how would/could the authors ensure the reliability of measurements after extrapolations, when the inflow rate varies within 50 l/min plus minus 5 and the same for the sampling time? Why did the sampling time differ?
Please add a sub-section of data analysis and statistics. Currently it is not clear what has been checked.
lines 224-229: I understand that the authors wish to compare the 'similarly acquired' results between the 2 different periods so as to identify 'long-term' differences, but this should, to my opinion, be a secondary only results of this paper. The paper deserves scientific merit, anyway. Also, as the authors did not (and cannot) provide detailed descriptions of the previously published results and methods, it is not easy for the reader to follow the concept and the comparability.
Figs. 3-4: please check axes titles for non-English language. Also, no error bars are provided. The authors state that they provide RMSD but are they so small that are not visible? All data refer to raw values right? And the analysis was a simple linear regression? The R square values are very high! From a check at the supplementary Tables, what kind of data did the authors use exactly? I saw most time "Sum of maximum values". Please provide a more detailed description on this. Also, in both Figs, the dotted lines are confidence intervals? It is indicated. To my opinion, it gets too complicated. If so, i would omit them.
Fig. 3: we cannot figure out the measurement units and years because of non-English.
Fig. 4: how would the authors explain the monthly variability? There are data at finer scales, right? Why not daily or weekly smoothed averaged values?
All figures, including Fig. 5: the authors do not state significance levels, only R square and equations. For instance, in Fig. 5, 2 of the 3 regression lines seem no-significant. Please provide the evidence throughout all Figs.
How do the authors explain the decreasing trend in Fig. 5?
Fig. 6 and overall comment for all Figs.: i am not sure if the authors need to present all non-significant results. Otherwise, i would prefer to see darker/solid lines those regressions that are significant, and lighter dotted lines without equations and R squares, the ones with no significant trends.
Fig. 6: i suppose no trend is significant?
Figs 6 and 7: it is not clear how are these indices elaborated and what their meaning exactly could be hear.
Fig. 8: also no trends. No need to show such Figs. Please focus on the significant results only.
Discussion, overall comment: The authors need to re-edit all the Results section, presenting the findings more appropriately, maybe also adding some more statistical analyses, focusing on statistically only results. Based on these, they need to edit from the scratch the whole Discussion. Currently is very short and superficial. Additional information on such biomonitoring techniques would be meaningful (airplane records and usefulness), a focus on the study areas and relation with extreme environments and climate change and comparisons of any robust trends against similar or different trends of similar bioaerosols in the world.
Conclusion: after modifying Results and Discussion, the authors should re-draw the main conclusions.
Overall comment: the English language needs thorough revision, preferably with the aid of a native speaker. There are repeated syntax and grammar errors throughout the manuscript.
Author Response
Author's Reply to the Review Report (Reviewer 2)
This is an important and novel contribution on long-term monitoring of bioaerosols in Siberia and relevant human health impacts. The subject as well as the study area are worth of investigation and hence the work deserves scientific merit. However, there are several issues that need to be resolved before publication, as follows:
Title: Based on many of the comments below, the authors need to change the title, taking care of terms like "long-term" (this usually refers to trends, sometimes of decadal scale), like "atmospheric bioaerosol" (see below), and "impacts on human health" (they do not really, this is only at a discussion level). I would recommend to completely change the phrasing.
The paper’s title was changed (lines 2 – 4) according to Reviewer’s recommendation.
line 17: please omit 'atmospheric': bioaerosols are by definition atmospheric. Please change throughout the manuscript.
I cannot do it in the first mention of the term. There different aerosol in occupational facilities, chambers, etc. After first mention I did it. It is written in some sentences as “bioaerosol” or “atmospheric aerosol” or even “aerosol”.
lines 18-19: not correct syntax, please edit.
The end of the Abstract is rewritten.
lines 21-23: as written here, it reads as if previously published results (of biomonitoring?) are compared against recently acquired experimental data, so it is immediately not clear whether indeed the 2 datasets are comparable. If indeed they are identical, please rephrase to something like 'a similar dataset was acquired using the exactly same methods as in... so as to check long-term alterations of/in...'.
The end of the Abstract is rewritten according to Reviewer’s remarks.
lines 25-26: please avoid using citations within the Abstract. Instead, use expressions like 'compared to previously published results by the authors in 2008', or similar.
The end of the Abstract is rewritten according to Reviewer’s remarks.
line 26: "set of indices" is too vague for the abstract. Please provide more details or name the most important.
The end of the Abstract is rewritten according to Reviewer’s remarks.
lines 28-30: there is an antiphasis here: "noticeable decrease" but "not changed significantly". Please provide statistical measures of both results, or change the phrasing.
The end of the Abstract is rewritten according to Reviewer’s remarks.
Overall comment in the Introduction: the Introduction is somewhat too technical, based on previously published results (and methods) by the authors. Before that, and so as to keep a balanced proportion of relevant information on the topic, i would expect additionally the state of the art on: A) monitoring of bioaerosols, with an emphasis at a 2nd stage on the selected microorganisms in this paper, B) previously published studies on environments like Siberia (extreme environmental regime, more sensitive to environmental stress), climate change effects, human health effects (and what kind). After those, the authors would make sense to highlight the novelty of their work: how the current research advances the scientific knowledge. My suggestion would be to focus more on the study area: Siberia alone is a great (and not that much published) environment, with higher sensitivity because of climate, latitude and extreme conditions. Discuss all accordingly.
New paragraph is added to INTRODUCTION. There are Siberian environment is discussed with new references added, health effect of Siberian environment with new references is added too (lines 83-96).
Methods, overall comment: the authors provide an extensive description of the processing/lab methods. However, the collection method is not proportionately described. Even if there is a previously published study, it is meaningful to briefly provide the concept of airplane measurements, as it is a very novel (and promising) biomonitoring tool. Please also add relevant information in the Introduction too. Moreover, add in Methods a more detailed description of the altitudes and the whole experimental design of the flight. A map would certainly significantly add up to the paper quality.
Information on collection methods is expanded (red font text lines 104-123), included information on aircraft sampling. The map is added as Figure 2 (in paper new version).
line 87: the differences are not really highlighted: if the reader has to go through all the methodological text to find these, it is confusing. Please create a dedicated sub-section highlighting potential differences, so as to make comparability clear and also ensure the soundness of the paper methods.
Sub-section 2.4 is added to test.
line 97: the devices used seem appropriate for this kind of sampling, but still, how have the authors ensured that all particles have been reliably and non-biased sampled under such a huge speed? Please discuss accordingly.
More information on impinger and intake device is added to the text (lines 113-123, red font). And photo of aircraft sampling stand with impingers is added too as Figure 4.
line 98: was the hour of the sampling taken into account? There is known diurnal periodicity of the bioaerosol abundance. Please discuss accordingly.
This information is added to sub-section 2.1.
line 98: how were the different altitudes taken into account? The sample was pooled across all elevations? A map would elucidate the issue.
No. Samples were studied for all altitudes separately. The information on altitudes time of sampling already exists in Tables S3, S6, S9, S12 and S13. This information partially added to sub-section 2.1.
lines 110-113: i totally understand the unavoidable variability of all measurements, especially with such a monitoring device, like an airplane. However, how would/could the authors ensure the reliability of measurements after extrapolations, when the inflow rate varies within 50 l/min plus minus 5 and the same for the sampling time? Why did the sampling time differ?
The random error of culture method is more than 30% usually. So, 10% random error of sampling rate is neglectable. 5 or 10 minutes is explained in lines 137-138.
Please add a sub-section of data analysis and statistics. Currently it is not clear what has been checked.
Sub-section 2.5 is added.
lines 224-229: I understand that the authors wish to compare the 'similarly acquired' results between the 2 different periods so as to identify 'long-term' differences, but this should, to my opinion, be a secondary only results of this paper. The paper deserves scientific merit, anyway. Also, as the authors did not (and cannot) provide detailed descriptions of the previously published results and methods, it is not easy for the reader to follow the concept and the comparability.
Agreed. It would be wonderful if reader can read our previous paper. It was cited many times in the text.
Figs. 3-4: please check axes titles for non-English language. Also, no error bars are provided. The authors state that they provide RMSD but are they so small that are not visible? All data refer to raw values right? And the analysis was a simple linear regression? The R square values are very high! From a check at the supplementary Tables, what kind of data did the authors use exactly? I saw most time "Sum of maximum values". Please provide a more detailed description on this. Also, in both Figs, the dotted lines are confidence intervals? It is indicated. To my opinion, it gets too complicated. If so, I would omit them.
Figs 5 and 6 (3 and 4 in previous version of the paper) are newly drawn. Dotted lines of the confidence intervals are omitted. See file Table S13: Integral index and data analysis in Supplementary Materials also.
Fig. 4: how would the authors explain the monthly variability? There are data at finer scales, right? Why not daily or weekly smoothed averaged values?
Fig. 6 (4 in previous version of the paper) is redrawn. Really samples were taken at VECTOR and AIRCRAFT site once a month, ss subsection 4.1. See comment in lines 406-413 also.
All figures, including Fig. 5: the authors do not state significance levels, only R square and equations. For instance, in Fig. 5, 2 of the 3 regression lines seem nonsignificant. Please provide the evidence throughout all Figs.
The Tables 1 - 4 are included in the text instead of omitted Figures 5 - 8. Data analysis is included in the Table S13 in Supplementary Materials.
How do the authors explain the decreasing trend in Fig. 5?
There is no reasonable explanation. It just happened at different sampling sites. Moreover, all these trends in Fig. 5 were undistinguishable from zero (see Table S13).
Fig. 6 and overall comment for all Figs.: i am not sure if the authors need to present all non-significant results. Otherwise, i would prefer to see darker/solid lines those regressions that are significant, and lighter dotted lines without equations and R squares, the ones with no significant trends.
The Tables 1 - 4 are included in the text instead of omitted Figures 5 - 8. Data analysis is included in the Table S13. All these trends in Fig. 6 were undistinguishable from zero (see Table S13).
Fig. 6: i suppose no trend is significant?
Yes, it is.
Figs 6 and 7: it is not clear how are these indices elaborated and what their meaning exactly could be hear.
The Tables 1 - 4 are included in the text instead of omitted Figures 5 - 8.
Fig. 8: also no trends. No need to show such Figs. Please focus on the significant results only.
The Tables 1 - 4 are included in the text instead of omitted Figures 5 - 8. All these trends in Fig. 6 were undistinguishable from zero (see Table S13).
Discussion, overall comment: The authors need to re-edit all the Results section, presenting the findings more appropriately, maybe also adding some more statistical analyses, focusing on statistically only results. Based on these, they need to edit from the scratch the whole Discussion. Currently is very short and superficial. Additional information on such biomonitoring techniques would be meaningful (airplane records and usefulness), a focus on the study areas and relation with extreme environments and climate change and comparisons of any robust trends against similar or different trends of similar bioaerosols in the world.
New text is added to DISCUSSION, lines 454-464.
Conclusion: after modifying Results and Discussion, the authors should re-draw the main conclusions.
New paragraph is added to CONCLUSION.
Overall comment: the English language needs thorough revision, preferably with the aid of a native speaker. There are repeated syntax and grammar errors throughout the manuscript.
Authors understand that even after correction English is not ideal. So, we are ready to present our paper for English polishing to Publisher. We understand that any work deserves fee. And we will be ready to pay for it.
And in the end we’d like to thank reviewer for his/her careful work. We believe that our paper will be much better due to his/her remarks. Thank you very much one more time!
Round 2
Reviewer 1 Report
I am happy with the revisions and recommend the manuscript for publishing. Some minor language polishing would probably still be required...
Author Response
Dear reviewer,
thank you very much for your work on improvement of our paper.
Reviewer 2 Report
The authors have followed the reviewers' comments and they have significantly improved their manuscript. Unfortunately, i believe they can further improve it, mainly having the following concerns:
Title:
Too long and complicated. Also, avoid pompous words like 'danger'. As highlighted in the previous review round, the authors have not really worked with clinical exposure or human health directly, so it is only based on a speculation level.
Abstract:
It has been greatly improved, but, as in the title, i would omit the term 'danger'. If the exposure is quantitative, the authors can use high exposure risk or increased abundances or similar.
lines 453-456: not necessarily: depends on the local vegetation, altitude, regional topography etc. Please discuss these.
The authors still lack discussing the novelty of their research in the frame of the unique study area. For instance, how future climate change could affect microbial composition and abundance in the air of Siberia? What could this mean for human health?
Also, the authors need to include a last paragraph in the Discussion about the limitations of the study and mainly about the sampling points that in some areas particularly (i.e. Kyuchi) may not reliably reflect the real microbial distribution/diversity/abundance. Please discuss this limitation and what the authors would expect if they had the whole seasonal diversity.
Regarding data analysis, more sophisticated analysis would provide more interesting results: ANOVA is nice for overall differences, but here the authors have repeated measures over time (within season and among years) and among 3 sites. An ANCOVA would elucidate non-significant trends or make the existing results more robust. My opinion is that the authors may have better results than they think.
Finally, there are still typos and some mistakes in English, but i understand their willingness to improve this at the final stage, after acceptance.
Author Response
Author's Reply to the Review Report (Reviewer 2)
The authors have followed the reviewers' comments and they have significantly improved their manuscript. Unfortunately, I believe they can further improve it, mainly having the following concerns:
Title:
Too long and complicated. Also, avoid pompous words like 'danger'. As highlighted in the previous review round, the authors have not really worked with clinical exposure or human health directly, so it is only based on a speculation level.
We change term “potential danger” to hazard in the Title, Abstract and in some phrases in the text (see red highlights).
Abstract:
It has been greatly improved, but, as in the title, i would omit the term 'danger'. If the exposure is quantitative, the authors can use high exposure risk or increased abundances or similar.
See comment above.
lines 453-456: not necessarily: depends on the local vegetation, altitude, regional topography etc. Please discuss these.
It is really so. But we estimating hazard for human in the sampling point and only for sampling period. The microorganisms’ concentration and their other properties depend on many factors. Some of these factors you mentioned above. Additional factors are: temperature, relative humidity, solar radiation, wind speed, air masses movement history, etc.
We changed lines 453-456 (now 451-453). We added some discussion in lines 463-485. And we added one new reference also.
The authors still lack discussing the novelty of their research in the frame of the unique study area. For instance, how future climate change could affect microbial composition and abundance in the air of Siberia? What could this mean for human health?
The paragraph (lines 486-490) was added to the text.
Also, the authors need to include a last paragraph in the Discussion about the limitations of the study and mainly about the sampling points that in some areas particularly (i.e. Kyuchi) may not reliably reflect the real microbial distribution/diversity/abundance. Please discuss this limitation and what the authors would expect if they had the whole seasonal diversity.
New paragraph (the last in Section 4) was added.
Regarding data analysis, more sophisticated analysis would provide more interesting results: ANOVA is nice for overall differences, but here the authors have repeated measures over time (within season and among years) and among 3 sites. An ANCOVA would elucidate non-significant trends or make the existing results more robust. My opinion is that the authors may have better results than they think.
We agree with this remark. It is possible that other methods of analysis would provide and interesting results. ANCOVA, unfortunately, does not. Due to its high variability and rather low number of culturable isolates in 2012-2016 trends calculated both ANOVA and ANCOVA are non-significant. Really our data are unique (without repeats) time series. We hope to use methods of time series analysis when we will have much more new data (experiments started in 2019). Right now, we think that ANOVA trends are enough for the paper. So, we did not change data analysis results in the text.
Really one of the paper’s aim is to remind readers about the method used showing new results. And to initiate researches start similar investigation somewhere else in the world. We hope that this additional aim is achieved.
Finally, there are still typos and some mistakes in English, but i understand their willingness to improve this at the final stage, after acceptance.
We confirm our willingness to improve this at the final stage, after acceptance.
And in the end we’d like to thank reviewer one more time for his/her work on improvement of our paper!
Round 3
Reviewer 2 Report
The authors have sincerely included the limitations of their study and they have already willingly applied most of the suggested changes. To my opinion, their manuscript is definitely novel and with high scientific standards and deserves to get published in the journal in its current form.
Only English corrections have to be made throughout, with the aid of the production Department.